# Fish heating tolerance scales similarly across individual physiology and populations

Nicholas L. Payne [1 ✉], Simon A. Morley [2], Lewis G. Halsey [3], James A. Smith[4], Rick Stuart-Smith[5], Conor Waldock [6,7] & Amanda E. Bates[8]

Extrapolating patterns from individuals to populations informs climate vulnerability models, yet biological responses to warming are uncertain at both levels. Here we contrast data on the heating tolerances of fishes from laboratory experiments with abundance patterns of wild populations. We find that heating tolerances in terms of individual physiologies in the lab and abundance in the wild decline with increasing temperature at the same rate. However, at a given acclimation temperature or optimum temperature, tropical individuals and populations have broader heating tolerances than temperate ones. These congruent relationships implicate a tight coupling between physiological and demographic processes underpinning macroecological patterns, and identify vulnerability in both temperate and tropical species.

[1] Trinity College Dublin, Dublin, Ireland. [2] British Antarctic Survey, Cambridge, UK. [3] University of Roehampton, London, UK. [4] Institute of Marine Sciences, University of California Santa Cruz, Santa Cruz, CA, USA. [5] Institute for Marine and Antarctic Studies, University of Tasmania, Nubeena Crescent, Taroona, TAS, Australia. [6] Landscape Ecology, Institute of Terrestrial Ecosystems, ETH Zurich, Zurich, Switzerland. [7] Swiss Federal Research Institute WSL, 8903 Birmensdorf, Switzerland. [8] Memorial University of Newfoundland, St. John's, NL, Canada. ✉email: paynen@tcd.ie

Research into the relationships between physiological limitations of species and their distributions has a rich history in ecology, formalized by the concept of fundamental versus realized niches[1]. Renewed interest is represented by the field of 'macrophysiology'[2], which explores large scale physiological variation and its ecological implications. Anthropogenic climate change has re-invigorated interest in thermal tolerances[3,4].

Comparing whether physiological functions of organisms and the performance of populations share responses to heating helps build a mechanistic understanding of heating responses across scales of biological organisation. However, relatively few generalizations can be made about patterns in the thermal tolerances of ectothermic animals[5,6]; less still about how tolerance patterns of individual organisms defined from laboratory experiments translate into those of populations in the wild[7,8]. Several comparative studies over the past decade have contrasted physiological thermal limits of species to the thermal limits of their geographical ranges, finding a number of interesting albeit complex patterns. For example, aquatic ectotherms appear to live closer to their physiological limits than do terrestrial species[9], whereas accounting for body temperature suggests terrestrial animals regularly exceed physiological thermal limits, necessitating behavioural thermoregulation[10]. Ectotherms from warmer climates tend to have higher physiological thermal tolerances (e.g. $CT_{max}$), but such limits are relatively conserved across lineages, possibly due to fixed physiological boundaries[11] or the non-linear increase in metabolism with temperature[12].

Since metrics of heating tolerance at different biological scales measure very different properties[13,14], the usefulness of laboratory-derived tolerance data for predicting climate change impacts is regularly questioned[8,15–17]. Nevertheless, understanding physiological variation via controlled experimentation could ultimately improve mechanistic understanding of ecological processes and therefore improve forecasts of species' responses to climate change[18,19]. One potential means of enhancing the applicability of physiological information for ecological predictions is to show that physiological patterns translate into ecological ones. The metabolic theory of ecology[20] is an influential example of macrophysiological patterns providing quantitative linkages between organismal, demographic and ecosystem processes, but similar frameworks are scarce for understanding how much heating organisms tolerate above temperatures to which they are acclimated or adapted.

We compared the temperature-dependence of physiological function in the laboratory with abundance in the wild by analysing data from two published datasets that together comprise more than 800 fish species. Our general aim was to explore whether there are similar patterns in fish thermal tolerance at the levels of individual physiology and abundance of natural populations. For physiological function of individual fish, we defined heating tolerance as the difference between the temperature a fish is acclimated to ($T_a$) and its upper critical thermal maximum in the laboratory ($CT_{max}$; measured as lethal temperatures or those coinciding with the loss of a critical physiological function; Fig. 1a). We called this 'physiological heating tolerance'. For wild abundance of fish populations, 'population heating tolerance' was defined as the difference between the temperature at which abundance is greatest ($T_{opt}$) and the temperature at the warm distribution limit of the species in its wild range ($T_{lim}$; Fig. 1b).

We found striking similarities between patterns in physiological heating tolerances in the lab and population heating tolerances in the wild, with both declining with increasing temperature at similar rates. We also found that, at a given acclimation temperature or optimum temperature, tropical individuals and populations have broader heating tolerances than

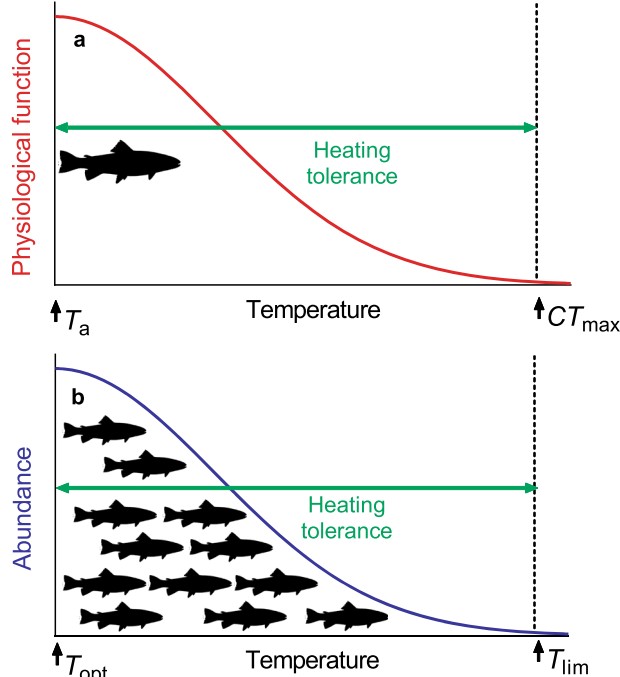

**Fig. 1 Conceptual illustration of fish heating tolerances at different biological scales. a** At the individual level, $T_a$ represents the temperature an individual is acclimated to in a laboratory experiment, and $CT_{max}$ is the temperature coinciding with death or loss of critical functions. **b** At the population level, $T_{opt}$ is the temperature of highest abundance in the wild, and $T_{lim}$ is the 95th percentile of maximum temperatures encountered by that species in its natural range.

temperate ones. These shared patterns at different biological scales suggest a potentially tight link between physiological and demographic processes, and highlight vulnerabilities in fishes from both temperate and tropical regions.

## Results
Physiological heating tolerances calculated from the laboratory estimates of critical performance are approximately two to three times higher than population heating tolerances calculated for abundance of wild fishes (Fig. 2a–b). This is not surprising given ectotherms are unlikely to live on the edge of their $CT_{max}$, and that heating tolerance decreases with slower rates of warming[21,22]. Indeed, individuals throughout a species' wild range tend to be exposed to changes in environmental temperatures over much longer time scales (months and years) than are those that undergo laboratory experimentation (days)[21]. It could perhaps be expected that the larger temperature fluctuations seen in the wild would elevate heating tolerances relative to those of fish held at relatively stable temperature regime in laboratory settings, but such an effect may simply buffer the larger difference between the highest temperature at which they can be competitive in nature ($T_{lim}$) and their physiological limit ($CT_{max}$). The difference between physiological and population heating tolerance is consistent with expectations that thermal range decreases for higher levels of biological complexity[14].

Both physiological and population heating tolerances strongly declined as ambient temperature increased (Fig. 2, Supplementary Tables 1–4; PGLS $P < 0.001$). That is, fishes acclimated to higher temperatures in the lab and wild fishes that were recorded at their highest densities at higher temperatures had lower heating tolerances (Fig. 2a–b). Moreover, the rates of

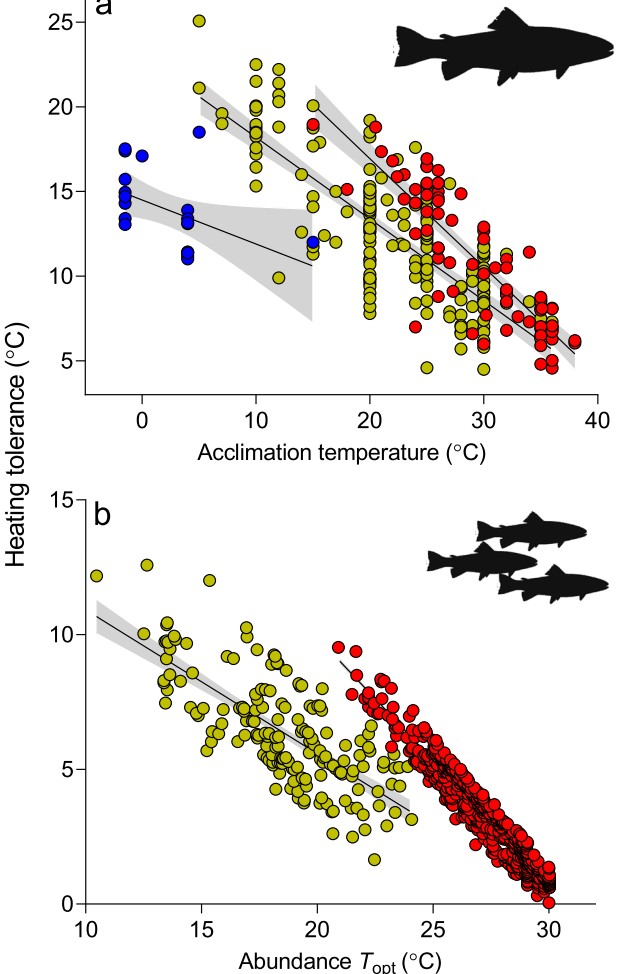

**Fig. 2 Fish heating tolerances at different biological scales. a** At the individual level, physiological heating tolerance is $CT_{max} - T_a$, and **b** at the population level, population heating tolerance is $T_{lim} - T_{opt}$ (as per Fig. 1). Tropical, temperate and polar species are indicated by red, yellow and blue, respectively. For panel **a**, each symbol represents a mean measurement ($n = 269$) from 121 different species, and for **b**, each symbol represents a different species ($n = 702$). Lines represent regression mean and 95% CIs.

these declines in heating tolerance with temperature were similar for individual physiology and wild populations (Fig. 2, Supplementary Tables 1–4).

Despite these similar, marked declines in heating tolerance toward higher temperatures, mean heating tolerances are broadly similar between temperate and tropical fishes at both physiological (12 vs 11 °C for temperate and tropical) and population (7 vs 3 °C) levels. As a result, because they generally experience warmer temperatures, tropical fishes tend to have larger heating tolerances than temperate ones at any given $T_a$ or $T_{opt}$. For example, a tropical fish population with $T_{opt}$ of 23 °C has roughly double the heating tolerance of a temperate species with the same $T_{opt}$ (Fig. 2b). Slopes were not significantly different between tropical and temperate species for physiological heating tolerance (PGLS interaction term $P = 0.196$, Supplementary Table 2), and only slightly steeper for tropical population heating tolerance (PGLS interaction term $P < 0.001$, Supplementary Table 4).

The similar rates of decline in heating tolerance across fish physiology and demography, with distinctive patterns for both tropical and temperate species, implicate a mechanistic link between these two biological scales. Whatever the ultimate mechanism

underpinning the decline in physiological heating tolerance at higher $T_a$ (and for tropical species to have higher heating tolerance at $T_a$ than temperate ones at the same $T_a$), it is possible that the physiological scaling patterns seen in Fig. 2a are regulating the range of temperatures that fish species encounter in the wild, in turn driving the abundance scaling patterns seen in Fig. 2b. The transition to higher heating tolerances in tropical species compared to temperate ones at a given $T_a$ or $T_{opt}$ does not seem readily explained by phylogenetic similarity, with closely related species spread across both guilds (Supplementary Fig. S1 and S2).

One possible explanation for why heating tolerances decline for warmer acclimated or adapted fishes, within either the temperate or tropical clades, is thermodynamics: 1 °C of heating tolerance represents a greater metabolic cost at higher $T_a$ or $T_{opt}$[12,23] because metabolism scales exponentially with temperature rather than linearly[24]. Alternatively, heating tolerance may decline toward firm upper limits of enzyme activity and/or environmental temperature limits available in the ocean (a particularly likely explanation for heating tolerances of the warmest-adapted tropical fish populations, which may also explain the slightly steeper slope in Fig. 2b)[25]. A more traditional explanation is that declining heating tolerances are an adaptive response to the more-stable environments toward warmer, tropical regions[26]. However, environmental temperature variability would not explain the transition to higher heating tolerances seen for the tropical guild of fish—'tropical' species ought to have lower heating tolerances than 'temperate' ones at the same $T_a$ or $T_{opt}$. Disentangling the influence of temperature and seasonality (i.e. latitude) is a classic biogeographer's problem, and our datasets imply that both factors may influence heating tolerance because heating tolerance declines with increasing temperature ($T_a$ and $T_{opt}$), but along different trajectories for tropical and temperate species.

The temperature an animal is acclimated to, $T_a$, is not necessarily the same as that species' "physiologically-optimal" temperature because physiological $T_{opt}$ can be defined in many different ways, and different processes are often maximised at different temperatures[16,27]. For example, Atlantic cod *Gadus morhua* have variously been measured as having $T_{opt}$ for oxygen supply (% venous $PO_2$) at 5 °C[28], $T_{opt}$ for aerobic scope being as low as 7 °C or higher than 14 °C[29,30], and $T_{opt}$ for growth rates ranging from 6 to 13 °C[31]. The acclimation process may improve performance at a given $T_a$, but it should be assumed neither that $T_a$ equates to $T_{opt}$ for all physiological processes, nor that physiological $T_{opt}$ approximates $T_{opt}$ for natural abundance. Our physiological and population heating tolerance datasets clearly measure different things, but their similar patterns are perhaps unsurprising if $T_{opt}$ for abundance is considered from the perspective of it being the temperature at which most individuals of a species are 'acclimated to' in the wild (i.e. the commonest $T_a$ in a species' range).

## Discussion

The similar scaling of patterns from the laboratory and the wild provides support for the application of across-species physiological trends to forecast how temperature governs distribution patterns (the basis of macrophysiology). Moreover, the observation of smaller heating tolerances at higher temperatures supports claims that ectotherms in warmer climates will be less resilient to future temperature rises[6]. Yet the transition to higher heating tolerances seen for tropical fishes than temperate species at the same $T_a$ or $T_{opt}$ (Fig. 2) suggests climate resilience assessments need to include more complexity than thermal regimes alone (at least for fishes). Studies like ours that combine physiological and ecological information could help explain how tropical species elevate their heating tolerance, a question of central importance for understanding future resilience.

## Methods

**Underlying data**. Data were derived from two recently published papers, with full physiological data compilation details and inclusion protocols found in Morley et al.[32], and wild abundance data and modelling approaches in Waldock et al.[7] Briefly, physiological data were compiled by Morley et al.[32] from a literature search of studies that tested the upper temperature limits (critical or lethal endpoints) of fish that were acclimated at more than one experimental temperature. The search terms "acclimate" or "acclimation" and "temperature" were used in Google, Google Scholar and Web of Knowledge, and latitude of specimen collection was used to delineate tropical, temperate and polar species (<30, 30–60, >60°, respectively). This search returned high and low $T_a$ measurements from 121 species (269 measurements in total) of juvenile and adult stages, from marine and freshwater habitats, and under experimental protocols that varied in some aspects between studies (e.g. rate of temperature ramping). These factors undoubtedly contribute to unexplained variation in our data. $T_{opt}$ and $T_{lim}$ were estimated by fitting quantile generalised additive models to species abundance data compiled by the global Reef Life Survey of shallow fish communities parameter estimates in ref. [7], raw abundance data presented in ref. [33]. These models related variation in abundance to thermal gradients, while accounting for confounding sources of environmental variation see ref. [7] for details.

$T_{lim}$ was estimated as the 97.5th quantile of minimum and maximum temperatures recorded at reef survey sites over a 2-year period[7], and tropical species were defined as those having $T_{lim} > 29$ °C. This division was chosen as it represents the threshold delineating the clear natural clustering of guilds (Fig. 2b). Waldock et al.[7] accounted for the effect of additional environmental variation such as $O_2$, phosphate, nitrate, current velocity, productivity, reef area, human population density, site depth, protection status scores and sampling intensity, and as such is robust to the confounding effects of environmental variation on thermal parameters. This dataset represented 702 species. Both datasets can be found in supporting online material in Morley et al.[32] and Waldock et al.[7], and are reproduced into a single file in supporting material of this paper.

**Statistics and reproducibility**. Analysis of both datasets was undertaken using the phylogenetic generalised least squares method, PGLS[34], with the 'caper' package[35] in R (version 3.3.0 R Foundation for Statistical Computing). Analyses that do not include phylogenetic information about the species represented treat each species as independent from all others. In reality, species are related and the covariance of traits exhibited by a species may be the result of its relatedness to other species with similar traits (phylogenetic inertia) rather than an instance of evolutionary adaptation see ref. [36] for illustration of the usefulness of PGLS methods. The fish phylogeny was based on a tree built using the 'rotl' package[37], which can be seen in Supplementary Fig. S1. A measure of phylogenetic correlation, $\lambda$ (lambda), was estimated by fitting PGLS models with different $\lambda$ values to uncover which value maximised the log likelihood. Lambda quantifies the degree to which trait evolution deviates from a 'Brownian motion' model (traits evolving by the accumulation of small, random changes over time)[38], and is thus considered to be a measure of the degree of phylogenetic correlation in the data[39]. $\lambda = 1$ retains the Brownian motion model, indicating that the trait covariance between any two species is directly proportional to their amount of shared evolutionary history, while $\lambda = 0$ indicates phylogenetic independence (the trait values across species are entirely unrelated to the phylogeny of those species). For the physiological dataset, we included in our models $T_a$, thermal guild (i.e. whether a species was derived from tropical, temperate or polar regions), whether the $T_a$ treatment was the higher or the lower for that species, and interactions between $T_a$ and those other two factors. For the population data, model terms were $T_{opt}$, thermal guild, and their interaction. We did not include the type of tolerance metric (e.g. critical or lethal endpoints) as a factor in our models because they are defined and measured in different ways, vary across taxa and the type of tolerance metric has been shown to vary neither by acclimation capacity[32] nor thermal tolerance breadth[9]. Alpha was set at 0.05 for all tests.

**Reporting summary**. Further information on research design is available in the Nature Research Reporting Summary linked to this article.

## Data availability

Data underlying the study can be found as Supplementary Data, and are available from paynen@tcd.ie upon request.

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

## Acknowledgements

We thank Andrew Jackson, Jacinta Kong and Craig White for thoughtful discussion of thermal tolerance. N.L.P. was supported by a Science Foundation Ireland Starting Investigator Research Grant (18/SIRG/5549).

## Author contributions

N.L.P., A.E.B. and S.A.M. conceived the study. L.G.H., J.A.S. and N.L.P. analysed the data. N.L.P. wrote the manuscript with contribution from S.A.M., L.G.H., J.A.S., R.S.S., C.W. and A.E.B.

## Competing interests

The authors declare no competing interests.
