## [Peer Review File · Communications Biology]

Reviewers' comments:

Reviewer #1 (Remarks to the Author):

Dear Authors

Congratulation for this great work.

After exhaustive revision of your MS entitled "Shared scaling of heating tolerance in fish individuals and populations" I have just 2 comments:

1.- In the figure 2A, the blue dots "Polar fish" appear a group with 15°C heating tolerance, it is not CTMax, which specie is it? and please put a short note about this.

2.- Line 54: you say: "Tcrit; measured as lethal temperatures or those coinciding with the loss of a critical physiological function" after many years working with fish you can have until 5 °C between LOE and lethal condition, my recommendation is to separate both condition or to clarify and to explain better the decision to keep it.

Reviewer #2 (Remarks to the Author):

Payne et al's manuscript "Shared scaling of heating tolerance in fish individuals and populations" provides a novel view on how the heating tolerance of fishes scales between two important levels of biological organization. Contrary to traditional views, the authors find interesting evidence that the heating tolerance of tropical fishes is not only comparable to temperate fishes, but even twice as high. The authors can rule out that phylogenetic similarity is driving these trends, and discuss potential thermodynamic and metabolic causes. The manuscript is well-written and highlights the importance of scaling physiological findings from the individual to the population-level to improve resilience assessments and ecosystem-based management approaches for fishes. Thus, Payne et al's findings are of immediate relevance for both a broader audience and the field of macrophysiology.

I have some more detailed comments, where I suggest the authors to clarify a few things regarding their methodologies and assumptions.

L1 If the conclusions were reached by using a data set on populations of shallow coral and rocky reef fishes the title should incorporate at least the term "reef". Otherwise, the implicit inclusion of highly mobile, pelagic species would be inadequate and an over-generalization of the findings.

L51 What data sets? Please add references or other details.

L53 The term critical thermal maximum has historically been associated with ramping assays to determine CTmax. The authors could consider using "critical upper thermal limits" to avoid confusion of the terms, or simply "upper temperature limits" (as used in L205).

L57-61 It would be helpful to outline that potentially confounding factors were accounted for in the analyses of Waldock et al 2019. Otherwise a reader may have reservations regarding the approach of relating temperature with (reef) fish abundances, as multiple environmental and anthropogenic drivers could theoretically contribute to the amount of observed fishes.

L61 It would be beneficial to include a statement here about why and how the comparative, phylogenetic approach was chosen and implemented before discussing the outcome in L93-97. I would even consider moving the supplementary figure representing the phylogenetic relationships to the main text. Please use a vector graphic for the phylogenetic tree to improve readability of species names inside the circles.

L190 Please add to the caption what each symbol shows (e.g. species), and what the regressions

and error bands represent (e.g. predicted means and SE).

L205 Why did the authors choose to include freshwater fishes when compiling the physiological data? If I understood it correctly, the population data were derived from marine coral and rocky reef fishes. Freshwater and estuarine fishes may experience stronger fluctuations in temperature, compared to fishes living on reefs. Thus, it might be more meaningful to compare marine species only.

L201-211 Please add further information how the physiological data were obtained and where they can be found. For example, consider explaining if the literature search was conducted to complement the dataset from Morley et al (2018), and provide the dataset online or in the supplementary material. Have you chosen a specific life stage, e.g. adults? Phenotypic, ontogenetic, and methodological differences can result in substantial intra-specific variability in upper thermal limits in fishes. Hence, it would be important to comment on how this was either accounted for or mention it as a shortcoming of the physiological data set. The collection and availability of the population data is sufficiently explained.

L217 It is not intuitive from Fig. 1b how the guilds are naturally clustered; consider providing a different example or remove the reference to the conceptual figure.

L222 Adding a sentence of why the PGLS procedure was chosen would help the reader understand the benefits of measuring the phylogenetic correlation when assessing important physiological traits, such as heat tolerance.

L233-237 I recommend adding further information on how the data were analysed (e.g. model selection processes and diagnostics including the relevant R packages). The additional information could be added to the supplementary information along with the model outcomes.

Reviewer #3 (Remarks to the Author):

Payne et al. have compiled an impressive and valuable dataset on the thermal tolerances of fish. While I can appreciate the effort required to compile this dataset, and the potential contribution it can make to our understanding of how climate can impact fish, I found myself somewhat confused as to the central finding of this paper and the key contribution that it made to the literature. I feel most (all?) of this confusion came from a lack of details in the paper, likely driven by an overly-succinct approach to writing. I also had several concerns related to methods and analyses that I feel need to be addressed. I therefore cannot recommend publication of this dataset without additional revisions. I hope my comments below will help this dataset move forward.

MAJOR COMMENTS

1. I struggled to extract the major contribution(s) and key findings of this study. After pouring through this document several times, I am fairly convinced I can identify the novelty and key finding of the study is, but I definitely had to work to have this finding squeak out. My sense is that the authors have skewed so heavily towards brevity with this document that they outsmarted themselves and did not adequately define the knowledge gap of this study and why the paper makes a large contribution to the literature. Therefore, at the very least, I feel the authors need to overhaul this manuscript to better define both the knowledge gap that this study fills, as well as the novelty of the findings and how this study moves this field forward. One suggestion would be to add additional text if needed. The current text is roughly half of what the journal will allow, and so there is a great deal of capacity to better place this study in the context of other work and drive home its key findings and significance.

2. Coupled with my previous comment, I felt that the methods of this paper were missing a number of details. It would appear that this study is coupled with previous ones where some of these details are outlined, but I felt this document, overall, was missing information. For example,

for the metric quantifying heating tolerance of individuals, the acclimation temperature of the fish was not shared. I realize that providing this information for 800 species would be burdensome, if not impossible. However, upper critical thermal maximum will correlate positively with acclimation temperature (<https://doi.org/10.1016/B978-0-12-374553-8.00200-8>), meaning that, within a species, animals acclimated to a lower temperature will have a lower T_{crit} relative to conspecifics acclimated to a higher T_{crit} . Similarly, fish are grouped into categories such as temperate or tropical based on the latitude of collection (Line 207), and additional details, citations or analyses need to be provided to prove the validity of these grouping based on these criteria.

3. Does the study need any kind of correction for phylogenetics/species? I realize that the supplementary file contains phylogenetic information, but it does not appear that this information was incorporated into models in any way. As one example, <https://doi.org/10.1111/gcb.13427> looked at how thermal tolerance differed across 82 freshwater fish species and noted a "strong phylogenetic signal in thermal tolerance . . . largely related to spatially autocorrelated adaptive processes" (page 732). The authors need to defend why phylogenetic information (e.g., phylogenetic contrasts, or similar methods of correcting for phylogeny) and/or autocorrelation, were not included in models as a fixed or random effect, ideally using peer-reviewed citations specific to this topic.

4. The dataset collected by Morley et al 2019 includes both marine and freshwater species from lab experiments, while the dataset generated by Waldock et al 2019 only includes marine species from field sampling. Although these are both fish species, in terms of environment variability, seasonal differences, and distribution limitations, freshwater species would be expected to experience greater temperature variation compared to marine species at the same latitude, which could influence thermal tolerance limits. The authors therefore need to defend the direct comparison of freshwater and marine fishes in this study and show that this comparison is valid and acceptable.

5. Many lab-based thermal tolerance studies arbitrarily choose acclimation temperatures for convenience, which can be seen in Figure 2 as there are a lot of data points at 20, 25, 30 degrees Celsius. As a result, the acclimation temperatures used in lab studies may not reflect the actual optimal temperatures for species in nature, which could affect their final heating tolerance performance. I am therefore concerned that the choice of lab acclimation temperatures could underestimate/overestimate thermal tolerance and cause skewed results in the comparison between tropical and temperate species, thus making lab-based and field-based dataset not comparable. The authors therefore need to clearly define why lab-based acclimation temperature was not included more prominently in analyses.

6. The comparisons between temperate and tropical species lack statistical analysis. For example, on Line 86 the authors indicate a "similar" rate of decline in heating tolerance, but there are no statistical tests to validate this comparison.

MINOR COMMENTS

7. Title: The used of 'shared' in title seems inappropriate. The word 'similar' would be a better word choice. In addition, the title seems vague and does not highlight the findings of the work, and therefore should be revised.

8. Lines 33-35: This section needs to be restructured to better emphasize how the response at the individual level can scale up to the population level.

9. Lines 52-56: using the term 'heating tolerance' as a term for two aspects of this study was somewhat confusing to me. In other words, the authors have 2 definitions of the term 'heating tolerance' (Line 52 and Line 56 of the manuscript). I can appreciate the simplicity of having only 1 term (especially in having a simple y-axis for Figure 2), but also think that the heating tolerance for an individual is very different than for a population. Therefore, the authors should consider changing one of these terms (even slightly) so they can be better differentiated to improve clarity.

10. Line 62-66: The authors need to consider that lab experiments normally hold fish at certain stable, fixed acclimation temperatures, while, in the wild, temperatures fluctuate almost all the time. This aspect needs to be better represented in the discussion.

11. Line 73: "ambient temperature increase" – change to "ambient temperature increases"

12. Line 94: "temperature ones" – change to "temperate ones"

13. Figures 1 and 2: Please ensure that font size is consistent across both figures.

Reviewer #1 (Remarks to the Author):

Dear Authors

Congratulation for this great work.

After exhaustive revision of your MS entitled "Shared scaling of heating tolerance in fish individuals and populations" I have just 2 comments:

Thanks for the positive summary.

1.- In the figure 2A, the blue dots "Polar fish" appear a group with 15°C heating tolerance, it is not CTMax, which specie is it? and please put a short note about this.

Yes the polar fish have heating tolerance ~ 15°C, which is similar to Tcrit because Ta are ~ 0 for most of these data. These polar species are from Nototheniidae, Zoarcidae, and Salmonidae. The full raw datasets including all species details will be uploaded when the paper is finally published.

2.- Line 54: you say: "Tcrit; measured as lethal temperatures or those coinciding with the loss of a critical physiological function" after many years working with fish you can have until 5 °C between LOE and lethal condition, my recommendation is to separate both condition or to clarify and to explain better the decision to keep it.

We have now added the following to clarify this point:

Ln 294-298: We didn't include the type of tolerance metric (e.g. critical or lethal endpoints) as a factor in our models because they are defined and measured in different ways, vary across taxa, and the type of tolerance metric has been shown to not vary by acclimation capacity (Morley et al., 2019) nor thermal tolerance breadth (Sunday et al., 2011).

Reviewer #2 (Remarks to the Author):

Payne et al's manuscript "Shared scaling of heating tolerance in fish individuals and populations" provides a novel view on how the heating tolerance of fishes scales between two important levels of biological organization. Contrary to traditional views, the authors find interesting evidence that the heating tolerance of tropical fishes is not only comparable to temperate fishes, but even twice as high. The authors can rule out that phylogenetic similarity is driving these trends, and discuss potential thermodynamic and metabolic causes. The manuscript is well-written and highlights the importance of scaling physiological findings from the individual to the population-level to improve resilience assessments and ecosystem-based management approaches for fishes. Thus, Payne et al's findings are of immediate relevance for both a broader audience and the field of macrophysiology. I have some more detailed comments, where I suggest the

authors to clarify a few things regarding their methodologies and assumptions.

Thanks for this positive appraisal. We have addressed each suggestion below.

L1 If the conclusions were reached by using a data set on populations of shallow coral and rocky reef fishes the title should incorporate at least the term “reef”. Otherwise, the implicit inclusion of highly mobile, pelagic species would be inadequate and an over-generalization of the findings.

We appreciate this rationale, however the physiological dataset is actually derived from a broader range of species than just shallow reef species (i.e. including freshwater, anadromous, polar etc) so we don't think the title is an over-generalisation.

L51 What data sets? Please add references or other details.

This has now been done.

L53 The term critical thermal maximum has historically been associated with ramping assays to determine CT_{max}. The authors could consider using “critical upper thermal limits” to avoid confusion of the terms, or simply “upper temperature limits” (as used in L205).

We have now replaced the term T_{crit} with CT_{max} throughout the ms as suggested (we prefer not to use the more general ‘upper temperature limits’ since this ms also deals with thermal limits of distribution, which could equally be considered ‘upper temperature limits’ and so might cause some confusion)

L57-61 It would be helpful to outline that potentially confounding factors were accounted for in the analyses of Waldock et al 2019. Otherwise a reader may have reservations regarding the approach of relating temperature with (reef) fish abundances, as multiple environmental and anthropogenic drivers could theoretically contribute to the amount of observed fishes.

We have now added sentences to the main text and methods to help ensure readers do not make the misinterpretation that the reviewer highlights:

Ln 81-83: “These models related variation in abundance to thermal gradients, whilst accounting for confounding sources of environmental variation (see Waldock et al., 2019 for details).”

Ln 262-266: “Waldock et al. (2019) accounted for the effect of additional environmental variation such as O₂, phosphate, nitrate, current velocity, productivity, reef area, human population density, site depth, protection status scores, and sampling intensity, and as such is robust to the confounding effects of environmental variation on thermal parameters.”

L61 It would be beneficial to include a statement here about why and how the comparative, phylogenetic approach was chosen and implemented before discussing the outcome in L93-97. I would even consider moving the supplementary figure representing the phylogenetic relationships to the main text. Please use a vector graphic for the phylogenetic tree to improve readability of species names inside the circles.

We have now added the following to the methods, as suggested, and have uploaded pdfs of the supplementary figures (we still think they are better there than in main text since they present information of secondary interest) to improve readability:

Ln 273-278: “Analyses that do not include phylogenetic information about the species represented treat each species as independent from all others. In reality, species are related and the covariance of traits exhibited by a species may be the result of its relatedness to other species with similar traits (phylogenetic inertia) rather than an instance of evolutionary adaptation (see Halsey et al., 2006 for illustration of the usefulness of PGLS methods)”

L190 Please add to the caption what each symbol shows (e.g. species), and what the regressions and error bands represent (e.g. predicted means and SE).

This has now been done.

L205 Why did the authors choose to include freshwater fishes when compiling the physiological data? If I understood it correctly, the population data were derived from marine coral and rocky reef fishes. Freshwater and estuarine fishes may experience stronger fluctuations in temperature, compared to fishes living on reefs. Thus, it might be more meaningful to compare marine species only.

We appreciate this point, but the objective of the ms is not to test how environmental variation influences heating tolerance (of individuals or populations). We agree that some freshwater habitats may encounter broader environmental temperature fluctuations than some marine habitats, but this is not always the case, and temperature fluctuations can also vary by latitude, by depth, by altitude, by season etc. Accordingly, we do not believe that excluding freshwater species on the basis of potential differences in temperature fluctuations is appropriate. Further, we do not wish to assume that temperature variation has a strong influence on acute heating tolerance to begin with – several papers that have explicitly tested this expectation have found temperature variation to have limited or no impact on thermal tolerance limits (e.g. Gunderson & Stillman 2015, Proc B).

L201-211 Please add further information how the physiological data were obtained and where they can be found. For example, consider explaining if the literature search was conducted to complement the dataset from Morley et al (2018), and provide the dataset online or in the supplementary material. Have you chosen a specific life stage, e.g. adults? Phenotypic, ontogenetic, and

methodological differences can result in substantial intra-specific variability in upper thermal limits in fishes. Hence, it would be important to comment on how this was either accounted for or mention it as a shortcoming of the physiological data set. The collection and availability of the population data is sufficiently explained.

We have now clarified that the lab data were collected in the earlier study, and have added the following to acknowledge some of the factors that probably contribute to variation that we do not account for:

Ln 251-266: This search returned high and low T_a measurements from 121 species (269 measurements in total) of juvenile and adult stages, from marine and freshwater habitats, and under experimental protocols that varied in some aspects between studies (e.g. rate of temperature ramping). These factors undoubtedly contribute to unexplained variation in our data.

We also added the following to clarify the location of the datasets:

Ln 267-269: Both datasets can be found in supporting online material in Morley *et al* (2019) and Waldock *et al* (2019), and are reproduced into a single file in supporting material of this paper.

L217 It is not intuitive from Fig. 1b how the guilds are naturally clustered; consider providing a different example or remove the reference to the conceptual figure.

Sorry this was a typo. We have amended this to refer to Fig 2b.

L222 Adding a sentence of why the PGLS procedure was chosen would help the reader understand the benefits of measuring the phylogenetic correlation when assessing important physiological traits, such as heat tolerance.

We have now added the following to the methods, as suggested to improve readability:

Ln 273-278: “Analyses that do not include phylogenetic information about the species represented treat each species as independent from all others. In reality, species are related and the covariance of traits exhibited by a species may be the result of its relatedness to other species with similar traits (phylogenetic inertia) rather than an instance of evolutionary adaptation (see Halsey et al., 2006 for illustration of the usefulness of PGLS methods)”

L233-237 I recommend adding further information on how the data were analysed (e.g. model selection processes and diagnostics including the relevant R packages). The additional information could be added to the supplementary information along with the model outcomes.

The following is included in the methods to capture this information:

Ln 271-286: Analysis of both datasets was undertaken using the phylogenetic generalised least squares method (PGLS; Grafen, 1989), with the caper package (Orme et al., 2013) in R (version 3.3.0 R Foundation for Statistical Computing). Analyses that do not include phylogenetic information about the species represented treat each species as independent from all others. In reality, species are related and the covariance of traits exhibited by a species may be the result of its relatedness to other species with similar traits (phylogenetic inertia) rather than an instance of evolutionary adaptation (see Halsey et al., 2006 for illustration of the usefulness of PGLS methods). The fish phylogeny was based on a tree built using the 'rotl' package (Michonneau et al., 2016), which can be seen in supplementary Fig 1. A measure of phylogenetic correlation, λ (lambda), was estimated by fitting PGLS models with different λ values to uncover which value maximised the log likelihood. Lambda quantifies the degree to which trait evolution deviates from a 'Brownian motion' model (traits evolving by the accumulation of small, random changes over time) (Freckleton, 2009), and is thus considered to be a measure of the degree of phylogenetic correlation in the data (Freckleton et al., 2002)

Reviewer #3 (Remarks to the Author):

Payne et al. have compiled an impressive and valuable dataset on the thermal tolerances of fish. While I can appreciate the effort required to compile this dataset, and the potential contribution it can make to our understanding of how climate can impact fish, I found myself somewhat confused as to the central finding of this paper and the key contribution that it made to the literature. I feel most (all?) of this confusion came from a lack of details in the paper, likely driven by an overly-succinct approach to writing. I also had several concerns related to methods and analyses that I feel need to be addressed. I therefore cannot recommend publication of this dataset without additional revisions. I hope my comments below will help this dataset move forward.

MAJOR COMMENTS

1. I struggled to extract the major contribution(s) and key findings of this study. After pouring through this document several times, I am fairly convinced I can identify the novelty and key finding of the study is, but I definitely had to work to have this finding squeak out. My sense is that the authors have skewed so heavily towards brevity with this document that they outsmarted themselves and did not adequately define the knowledge gap of this study and why the paper makes a large contribution to the literature. Therefore, at the very least, I feel the authors need to overhaul this manuscript to better define both the knowledge gap that this study fills, as well as the novelty of the findings and how this study moves this field forward. One suggestion would be to add additional text if needed. The current text is roughly half of what the journal will allow, and so there is a great deal of

capacity to better place this study in the context of other work and drive home its key findings and significance.

In line with this comment, we have now expanded the main text by ~ 50%, with a view to better contextualise the work and explain its significance. We feel this improves readability, and more-clearly identifies the knowledge gap. The following additional text has been added:

Ln 41-51: Several comparative studies over the past decade have contrasted physiological thermal limits of species to the thermal limits of their geographical ranges, finding a number of interesting albeit complex patterns. For example, aquatic ectotherms appear to live closer to their physiological limits than do terrestrial species (Sunday et al., 2011), whereas accounting for terrestrial body temperature suggests those animals regularly exceed physiological thermal limits, necessitating behavioural thermoregulation (Sunday et al., 2014). Ectotherms from warmer climates tend to have higher physiological thermal tolerances (e.g. CT_{max}), but such limits are relatively conserved across lineages, possibly due to fixed physiological boundaries (Araujo et al., 2013), or the non-linear increase in metabolism with temperature (Payne & Smith, 2017).

Ln 52-61: Since metrics of heating tolerance at different biological scales measure very different properties (Rezende & Bozinovic, 2019; Sinclair et al., 2016), the usefulness of laboratory-derived tolerance data for predicting climate change impacts in nature is regularly questioned (Barnes et al., 2010; Clark et al., 2013; Norin et al., 2014; Payne et al., 2016). Nevertheless, understanding physiological variation *via* controlled experimentation could ultimately improve mechanistic understanding of ecological processes and therefore improve forecasts of species' responses to climate change (Buckley et al., 2010; Kearney & Porter, 2009). One potential means of enhancing the applicability of physiological information for ecological predictions is to show that physiological patterns translate into ecological ones.

Ln 68-70: Our general aim was to explore whether there are similar patterns in fish thermal tolerance at the level of individual physiology and abundance of natural populations

Ln 146-160: The temperature an animal is acclimated to, T_a , is not necessarily the same as that species' "physiologically-optimal" temperature because physiological T_{opt} can be defined in many different ways, and different processes are often maximised at different temperatures (Clark et al., 2013; Martin & Huey, 2008). For example, Atlantic cod *Gadus morhua* have variously been measured as having T_{opt} for oxygen supply (% venous PO_2) at 5°C (Pörtner et al., 2008), T_{opt} for aerobic scope being as low as 7°C, or higher than 14°C (Claireaux et al., 2000; Sylvestre et al., 2007), and T_{opt} for growth rates ranging from 6 to 13°C (Björnsson & Steinarsson, 2002). The acclimation process may improve performance at a given T_a , but it should not be assumed that T_a equates to T_{opt} for all physiological processes, nor that physiological T_{opt} approximates T_{opt} for natural abundance. Our physiological and population heating tolerance datasets clearly measure different things, but their similar patterns are perhaps unsurprising if T_{opt} for abundance is

considered from the perspective as the temperature at which most individuals of a species are 'acclimated to' in the wild (i.e. the commonest T_a in a species' range).

2. Coupled with my previous comment, I felt that the methods of this paper were missing a number of details. It would appear that this study is coupled with previous ones where some of these details are outlined, but I felt this document, overall, was missing information. For example, for the metric quantifying heating tolerance of individuals, the acclimation temperature of the fish was not shared. I realize that providing this information for 800 species would be burdensome, if not impossible. However, upper critical thermal maximum will correlate positively with acclimation temperature (<https://doi.org/10.1016/B978-0-12-374553-8.00200-8>), meaning that, within a species, animals acclimated to a lower temperature will have a lower T_{crit} relative to conspecifics acclimated to a higher T_{crit} . Similarly, fish are grouped into categories such as temperate or tropical based on the latitude of collection (Line 207), and additional details, citations or analyses need to be provided to prove the validity of these grouping based on these criteria.

The acclimation temperature of each data point is in fact represented by the x-axis in Fig 2a. We agree with the summary of how T_a influences T_{crit} , and indeed that is a key focus of our manuscript (specifically, how much does T_{crit} increase for a given increase in T_a ; this is the T_a -HT relationship in Figure 2a). We do not agree that citations or additional analyses are required to justify using latitude to classify fish as tropical, temperate or polar; those are standard terms used to describe latitudinal categories.

3. Does the study need any kind of correction for phylogenetics/species? I realize that the supplementary file contains phylogenetic information, but it does not appear that this information was incorporated into models in any way. As one example, <https://doi.org/10.1111/gcb.13427> looked at how thermal tolerance differed across 82 freshwater fish species and noted a "strong phylogenetic signal in thermal tolerance . . . largely related to spatially autocorrelated adaptive processes" (page 732). The authors need to defend why phylogenetic information (e.g., phylogenetic contrasts, or similar methods of correcting for phylogeny) and/or autocorrelation, were not included in models as a fixed or random effect, ideally using peer-reviewed citations specific to this topic.

In line with this comment and a similar one from R2, we have now added the following to explain why our phylogenetic approach was taken. As suggested, we now include reference to a paper that discusses this analysis in more detail

Ln 273-278: Analyses that do not include phylogenetic information about the species represented treat each species as independent from all others. In reality, species are related and the covariance of traits exhibited by a species may be the result of its relatedness to other species with similar traits (phylogenetic inertia) rather than an

instance of evolutionary adaptation (see Halsey et al., 2006 for illustration of the usefulness of PGLS methods)

4. The dataset collected by Morley et al 2019 includes both marine and freshwater species from lab experiments, while the dataset generated by Waldock et al 2019 only includes marine species from field sampling. Although these are both fish species, in terms of environment variability, seasonal differences, and distribution limitations, freshwater species would be expected to experience greater temperature variation compared to marine species at the same latitude, which could influence thermal tolerance limits. The authors therefore need to defend the direct comparison of freshwater and marine fishes in this study and show that this comparison is valid and acceptable.

We appreciate this point, but the objective of the ms is not to test how environmental variation influences heating tolerance (of individuals or populations). We agree that some freshwater habitats may encounter broader environmental temperature fluctuations than some marine habitats, but this is not always the case, and temperature fluctuations can also vary by latitude, by depth, by altitude, by season etc. Accordingly, we do not believe that excluding freshwater species on the basis of potential differences in temperature fluctuations is appropriate. Further, we do not wish to assume that temperature variation has a strong influence on acute heating tolerance to begin with – several papers that have explicitly tested this expectation have found temperature variation to have limited or no impact on thermal tolerance limits (e.g. Gunderson & Stillman 2015, Proc B).

5. Many lab-based thermal tolerance studies arbitrarily choose acclimation temperatures for convenience, which can be seen in Figure 2 as there are a lot of data points at 20, 25, 30 degrees Celsius. As a result, the acclimation temperatures used in lab studies may not reflect the actual optimal temperatures for species in nature, which could affect their final heating tolerance performance. I am therefore concerned that the choice of lab acclimation temperatures could underestimate/overestimate thermal tolerance and cause skewed results in the comparison between tropical and temperate species, thus making lab-based and field-based dataset not comparable. The authors therefore need to clearly define why lab-based acclimation temperature was not included more prominently in analyses.

We actually think lab-based acclimation temperature, T_a , was included quite prominently in the analyses. Indeed, that is the focus of Fig 2a, the conceptual Figure, and much of our discussion. We agree that the relationship between T_a and T_{opt} could have been explained a little more clearly, so we have now added the following paragraph to the discussion:

Ln 146-160: The temperature an animal is acclimated to, T_a , is not necessarily the same as that species' "physiologically-optimal" temperature because physiological T_{opt} can be defined in many different ways, and different processes are often maximised at different temperatures (Clark et al., 2013; Martin & Huey, 2008). For example, Atlantic cod *Gadus morhua* have variously been measured as having T_{opt} for oxygen supply (% venous PO_2) at 5°C (Pörtner et al., 2008), T_{opt} for aerobic

scope being as low as 7°C, or higher than 14°C (Claireaux et al., 2000; Sylvestre et al., 2007), and T_{opt} for growth rates ranging from 6 to 13°C (Björnsson & Steinarsson, 2002). The acclimation process may improve performance at a given T_a , but it should not be assumed that T_a equates to T_{opt} for all physiological processes, nor that physiological T_{opt} approximates T_{opt} for natural abundance. Our physiological and population heating tolerance datasets clearly measure different things, but their similar patterns are perhaps unsurprising if T_{opt} for abundance is considered from the perspective as the temperature at which most individuals of a species are 'acclimated to' in the wild (i.e. the commonest T_a in a species' range).

6. The comparisons between temperate and tropical species lack statistical analysis. For example, on Line 86 the authors indicate a “similar” rate of decline in heating tolerance, but there are no statistical tests to validate this comparison.

We did in fact include interaction terms in our analyses to test whether slopes varied by thermal guild (e.g. tropical vs temperate); these results were included in the supplementary table. We have now also clarified this point in the main text:

Ln 113-116: Slopes were not significantly different between tropical and temperate species for individual heating tolerance (supplementary Tables 1 and 2), and only slightly steeper for tropical heating tolerance of populations (supplementary Tables 1 and 2).”

We do not formally (statistically) compare slopes between our individual and population level datasets given they are different datasets that measure different things.

MINOR COMMENTS

7. Title: The used of ‘shared’ in title seems inappropriate. The word ‘similar’ would be a better word choice. In addition, the title seems vague and does not highlight the findings of the work, and therefore should be revised.

As suggested, we have now changed the title to

“Similar scaling of fish heating tolerance across individual physiology and natural abundance”

8. Lines 33-35: This section needs to be restructured to better emphasize how the response at the individual level can scale up to the population level.

Following this Reviewer’s main comment, we have now added a further ~ page of text to better explain this point.

9. Lines 52-56: using the term ‘heating tolerance’ as a term for two aspects of this study was somewhat confusing to me. In other words, the authors have 2 definitions of the term ‘heating tolerance’ (Line 52 and Line 56 of the manuscript). I can appreciate the simplicity of having only 1 term (especially in

having a simple y-axis for Figure 2), but also think that the heating tolerance for an individual is very different than for a population. Therefore, the authors should consider changing one of these terms (even slightly) so they can be better differentiated to improve clarity.

We have now defined the two datasets as 'physiological heating tolerance' and 'population heating tolerance' (Ln 74-75), and taken more care to use this consistent terminology throughout the ms.

10. Line 62-66: The authors need to consider that lab experiments normally hold fish at certain stable, fixed acclimation temperatures, while, in the wild, temperatures fluctuate almost all the time. This aspect needs to be better represented in the discussion.

We have now added the following text to reflect this point:

Ln 92-96: It could perhaps be expected that the larger temperature fluctuations seen in the wild would elevate heating tolerances relative to those of fish held at relatively-stable temperature regime in laboratory settings, but such an effect may simply buffer the larger difference between the highest temperature at which they can be competitive in nature (T_{lim}) and their physiological limit (CT_{max}).

11. Line 73: “ambient temperature increase” – change to “ambient temperature increases”

This typo has now been fixed.

12. Line 94: “temperature ones” – change to “temperate ones”

This typo has now been fixed.

13. Figures 1 and 2: Please ensure that font size is consistent across both figures.

We have now increased font size for Fig 2. It remains a little smaller than Fig 1 because we anticipate that figure to be larger in print.

REVIEWERS' COMMENTS:

Reviewer #1 (Remarks to the Author):

Dear Author

I'm agree with yours revisions and for me the ms can be accepted.

Reviewer #2 (Remarks to the Author):

Payne and colleagues have made substantial improvements to their revised manuscript, now titled "Similar scaling of fish heating tolerance across individual physiology and natural abundance". The authors now provide a better overview of the topic and a rationale for the study. Similarly, the information added to the methods section (e.g. on the used data set) makes it easier for the reader to understand the analyses. Using standard terminology, i.e. CTmax, and more descriptive terms, such as "physiological heating tolerance" and "population heating tolerance", improved the text as well. The edits made to the discussion now include potential shortcomings (e.g. L149-163) allowing the reader to interpret the findings more objectively. In summary, the authors have addressed the reviewers' comments well and significantly improved the manuscript.

Reviewer #3 (Remarks to the Author):

The authors have done an excellent job addressing my comments. I have no further suggestions or questions.